# Assessment of the Attraction Range of Sex Pheromone Traps to *Agriotes* (Coleoptera, Elateridae) Male Click Beetles in South-Eastern Europe

**DOI:** 10.3390/insects12080733

**Published:** 2021-08-16

**Authors:** Lorenzo Furlan, Barbara Contiero, Miklós Tóth

**Affiliations:** 1Agricultural Research Sector, Veneto Agricoltura, 35020 Legnaro, Italy; 2Department of Animal Medicine, Production and Health, University of Padua, 35020 Legnaro, Italy; barbara.contiero@unipd.it; 3Plant Protection Institute, Centre for Agricultural Research, P.O. Box 102, H-1525 Budapest, Hungary; toth.miklos@atk.hu

**Keywords:** sampling area, wireworms, *Agriotes brevis*, *A. litigiosus*, *A. sordidus*, *A. ustulatus*, IPM

## Abstract

**Simple Summary:**

Click beetles are the adults of wireworms, soil-dwelling larvae that damage multiple arable crops. The attraction range of YATLORf pheromone traps to click beetles of four species of *Agriotes* wireworms was studied to provide additional information about the implementation of integrated pest management against these harmful pests in Europe. This should allow a significant reduction in insecticide use. Male click beetles were marked and released at different distances from a pheromone trap. Recapture rate, maximum sampling ranges, and effective sampling areas were calculated. The recapture rate was significantly affected by distance, species, and wind direction and decreased as distance increased. The majority of beetles were caught from short distances (up to 10 m) within the first five days. The estimated attraction range was low for all the considered *Agriotes* species, suggesting that pheromone traps were unsuitable for use as mass trapping instruments to disrupt mating. However, after results and previous research outputs were evaluated, it seems possible to use the traps not only as monitoring tools, but also as attract-and-kill strategies for most beetle populations.

**Abstract:**

The attraction range of YATLORf pheromone traps to adults of four species of *Agriotes* (*A. brevis*, *A. sordidus*, *A. litigiosus*, and *A. ustulatus*) was studied to provide additional information about the most harmful *Agriotes* species in Europe. Male click beetles were marked and released at different distances from a pheromone trap. The recapture rate was calculated and analyzed using analysis of variance. The recapture rate was significantly affected by distance, species, and wind direction. The recapture rate decreased as distance increased. The majority of beetles were caught from short distances (up to 10 m) within the first five days. *A. brevis*, a mainly crawling species, showed the lowest recapture rate. The wind direction affected the recovery rate, with a significantly lower number of beetles moving downwind from the release points. Maximum sampling ranges and effective sampling areas were calculated. The obtained estimations were low (53 to 86 m and 509 to 2602 m^2^, respectively) for all the considered *Agriotes* species, suggesting that they were unsuitable for use as mass trapping instruments to disrupt mating. However, it seems possible to use the traps not only as monitoring tools, but also as attract-and-kill strategies for most beetle populations.

## 1. Introduction

Wireworms, the larvae of click beetles (Coleoptera: Elateridae), rank among the main soil pests of several arable crops (e.g., maize, winter wheat) in Europe and North America [1]. A number of crop-damaging click beetle species are widespread in Europe: *A. brevis* Candeze, *A. lineatus* L., *A. litigiosus* Rossi, *A. obscurus* L., *A. proximus* Schwarz, *A. rufipalpis* Brullé, *A. sordidus* Illiger, *A. sputator* L., and *A. ustulatus* Schäller [2]. *A. brevis*, *A. litigiosus*, *A. sordidus,* and *A. ustulatus* are the major crop-damaging species in Italy’s Po Valley [3,4]. *A. sordidus* has become a serious pest in Germany [5] after being reported as a main pest in France [6]; long-term research in Northeast Italy found this species to be the most frequent cause of severe damage to maize, alongside *A. brevis* [7]. *A. brevis* is a major pest in Italy, as well as in Eastern European countries [7,8,9,10,11,12,13,14]. *A. ustulatus* is also widespread and causes damage in Central and Eastern Europe [2,5,8,9,10,12,15,16,17], while *A. litigiosus* is important in Italy, Greece, and several Eastern European countries [9,10,18,19,20].

*Agriotes* species have a similar life cycle, featuring a prolonged period spent as larvae in the soil before pupation. They can be divided into two groups: species overwintering as adults and species not overwintering as adults [21]; *A. brevis*, *A. lineatus*, *A. obscurus*, *A. proximus*, *A. rufipalpis*, *A. sordidus,* and *A. sputator* belong to the former group, while *A. ustulatus* and *A. litigiosus* belong to the latter. 

Life-cycle duration is generally 2–3 years [22,23,24], with only the adult stage dwelling outside the soil; a few days for species nonoverwintering as adults [15,21,22], and many months for species overwintering as adults [23,24]. 

Since monitoring insects in the soil is difficult and expensive, efforts have been made to assess population levels of the only stage living outside the soil: adults. The female sex pheromones of most major European click beetle pest species (*A. brevis*, *A. lineatus*, *A. obscurus*, *A. proximus*, *A. rufipalpis*, *A. sordidus*, *A. sputator*, *A. ustulatus*, * A. litigiosus*) have been characterized [25]. YATLORf (Yf) sex pheromone traps have been designed for a range of *Agriotes* species [9,25]. They have proven to be highly efficient in capturing click beetle (*Agriotes*) species [9], and a clear relationship was found between male click beetle catches in pheromone traps and subsequent wireworm abundance in the related area for at least three species: *A. brevis*, *A. sordidus* and *A. ustulatus* [26]. This made it possible to predict potential wireworm damage, thus improving integrated pest management (IPM) of wireworms. 

After Yf sex pheromone traps had been found to be an effective monitoring tool, there was increased interest in understanding the potential of trap catches for IPM, and particularly their actual attraction range. The first published study concerned three species overwintering as adults: *A. obscurus*, *A. lineatus* and *A. sputator*. Hicks and Blackshaw [27] noted a significant decrease in the number of *A. lineatus* recaptured by pheromone traps from a 16 m release distance (42.2%) compared with 4 m (91.7%). Similarly, a clear decrease in the recapture rate with increasing release distance (4 m vs. 16 m) was found for *A. obscurus* (75% vs. 29.8%) and *A. sputator* (33.3% vs. 2.2%). Sufyan et al. [28] carried out a longer, more complete study, concluding that the attraction range of pheromone traps to *A. lineatus* and *A. obscurus* is comparatively low, with traps recapturing high rates of beetles from release distances up to 10 m only. Under field conditions, the spatial distribution of naturally occurring male click beetles strongly affects the outcome of pheromone male trapping [29]. In this study, we used the mark–release–recapture approach to investigate the attraction range of pheromone traps to *A. brevis*, *A. sordidus*, *A. litigiosus,* and *A. ustulatus* in order to extend information about the potential of Yf sex pheromone traps for IPM, including the prediction of crop damage risk and the feasibility of mass trapping and mating disruption, plus attract-and-kill strategies.

## 2. Materials and Methods

The general experimental design was the same as that followed by Sufyan et al. [28] and, with some modifications, Hick and Blackshaw [27]. 

### 2.1. Location

Experiments were carried out at farms in Venice Province, Veneto, Italy (San Donà di Piave 45.640915, 12.587629 and Eraclea 45.611825, 12.661857) from 2000 to 2003 on different types of ground cover (maize, soybean, sugar beet, winter wheat, meadow, and bare soil). They took place during the natural dispersal peaks of each species [15,21,22,23] (see Table 1).

### 2.2. Sex Pheromone Traps

Yf traps [9,30] produced by ROSA Micro S.r.l. (Ceggia, Venice, Veneto, Italy) were used. The pheromone lure dispensers are commercially obtainable from the CSALOMON^®^ trap family (Plant Protection Institute, Budapest, Hungary). Compositions of single lures comprised *Agriotes brevis* geranyl butanoate + (E,E)-farnesyl butanoate 1:1 (15 + 15 mg, [31]), *Agriotes sordidus* Illiger geranyl hexanoate 30 mg [32], *Agriotes litigiosus* geranyl isovalerate 50 mg, and *Agriotes ustulatus* (E,E)-farnesyl acetate 50 mg [25]. The Yf trap’s white bottom was placed facing down, with its brown edge 1–2 cm below the soil. We used two lure positions: a low position for *A. brevis* and *A. sordidus*, and a high position for *A. litigiosus* and *A. ustulatus*, with the lure topside facing down in both cases (Figure 1 [26]). On every inspection, the bottom was cleaned up, with soil and any residuals being removed. The insects were removed from the trap as follows. The trap was removed from the soil; it was then placed inside a large plastic bag, which was opened so that the insects dropped inside, and the bag was closed immediately after the trap had been removed. The trap was then returned to its initial position. All the individuals were preserved in cool conditions (5–8 °C) for taxonomic identification [33].

### 2.3. Beetle Collection, Marking and Release

Adult click beetles (*A. brevis*, *A. litigiosus*, *A. sordidus,* and *A. ustulatus*) were captured using Yf sex pheromone traps [9,26] from a number of infested fields between the beginning and the peak of their swarming period: late March to April for *A. brevis*; late April to May for *A. sordidus* [23]; late May to June for *A. litigiosus* [18]; early June to mid-July for *A. ustulatus* [15]. These traps proved to be suitable for catching both flying and crawling species throughout the season. All captured beetles were sexed and identified to species [33], with the males placed in aerated boxes containing moist soil and fresh Gramineae leaves until the field experimental treatments were established (maximum six days). 

Different Uni Posca^®^ waterproof colors were painted onto the elytra of the male beetles. Six main colors were combined with five beetle-spotting positions: prothorax, right elytra front, left elytra front, right elytra back, and left elytra back. This wide range of unique combinations enabled each of the 12/24 treatments to be identified easily. There were six release distances: 2, 5, 10, 15, 20, and 60 m, with two or four release directions. The beetles were either released east (upwind) of the trap, with them moving downwind towards the pheromone source (prevailing east wind), or west (downwind) of the trap, with them moving upwind towards the pheromone source (wind likely taking the pheromone plumes to the beetles). North and south release directions were added in some trials. Each release direction was a replication. The closest weather station (within 5 km) and local observation were used to confirm the wind directions post-experiment. All the trials were carried out under a calm-to-light east breeze (<6 km/h) on the Beaufort Wind Force Scale, at least for the first three days. In most cases, the color markings of the recaptured males were visible to the naked eye. When color recognition was difficult, binoculars (up to 100× magnitude) were used. One Yf pheromone trap per trial was placed in the selected field with no wind obstacles. In most cases, 25 beetles of each species were marked and released at each point (Table 1), although numbers ranged from 10 to 100. Before release, the beetles were transported to the experimental site in small vials and allowed to leave the vial on their own. One vial containing the selected beetles was placed at each release distance from the trap and rapidly opened. Vials were placed rapidly, between one and two minutes, starting from the furthest distances. Once the beetles had been released, the pheromone traps were inspected after 10 min, 1 h, and 1, 3, 5, 12, and 30–36 days. Both marked and unmarked trapped beetles were collected for identification [33] and counted. Only marked beetles were considered for the statistical analyses. The recapture rate for males was calculated as: recaptured/released × 100. Approximately 20% of the males never left the release point as they were probably too weak to fly, so the recapture rate was adjusted to take account of these no-releases [27]. The experiment was repeated at different trial locations at least 300 m apart. Trials with the same species were carried out 4–6 kilometers apart and at intervals of several days. 

### 2.4. Statistical Analysis 

The total recapture rate per species was evaluated by analysis of variance using SAS, version 9.4 (SAS Institute, Cary, NC, USA). The model included the fixed effects of species, distance (six release distances: 2, 5, 10, 15, 20, and 60 m) and direction (north and east (downwind); south and west (upwind)), using the trials as replications. Precipitation (mm) and mean temperature were included in the model as covariates. Tukey’s test was used for post hoc comparisons of the means. The hypotheses of linear model on the residuals (normality, independence, and homoscedasticity) were graphically assessed. For the calculation of maximum range and sampling area, Sufyan et al. [28] used the same statistical method as Turchin and Odendaal [34], while Hicks and Blackshaw [27] used the one suggested by Östrand and Anderbrant [35], which was a modified version of the Turchin and Odendaal [34] method. The present work adopted the method used by Sufyan et al. [28], which is very similar to the method used by Hicks and Blackshaw [27]. This choice of method allowed us to measure the reactions of all the main *Agriotes* species to pheromone traps (range of attraction) and enabled our findings to be compared quickly and simply with previous ones. The quantitative relationship between P(r) (probability of recapture, percentage of beetles recaptured), considered as dependent variable y, and r (release distance), considered as predictive-independent variable x, was represented by a linear model: P(r) = a + b × r [34]. The maximum sampling range (r_s_) was calculated by solving for P(r) = 0 in the linear model. This means that r_s_ = −a/b, where a is the intercept of the linear model, and b is the regression coefficient. Different linear regression models using untransformed or transformed data (base 10 logarithmic transformation) were employed to calculate the effective sampling area (α), selecting the relationship with the highest R^2^. The relationship between r and log P(r) showed the highest averaged R^2^ over species and time periods, and it was used to calculate the sampling area correspondent to the maximum sampling range. Due to the assumed relation between P(r) and r, we obtained:log10 P(r) = a + b × r which means P(r) = 10^(a + b × r)^

By substituting this relation in the Turchin and Odendaal [34] equation, we obtained
$$\alpha =2\pi \int_{0}^{r_{s}}r\times P\left(r\right)\mathrm{dr}\;=2\pi \int_{0}^{r_{s}}r\;10^{a+b\;r}\mathrm{dr}$$
where α is the sampling area. 

The integrals were calculated using Wolfram Mathematica (online calculator, http://integrals.wolfram.com/index.jsp, accessed date: 31 January 2021).

## 3. Results

All of the statistical analysis focuses on records up to day 12, because after this time only three beetles were recaptured. The total cumulative recapture rate ranged from 19% for *A. brevis* and *A. ustulatus*, 26% for *A. litigiosus*, and 30% for *A. sordidus*. By day 12, more than 50% of the *A. sordidus* beetles released 2 m from the trap had been recaptured, whereas only 30% of *A. litigiosus* had been recaptured (Figure 2a). Most of the beetles were recaptured within five days after release, regardless of distance, except for the *A. brevis* beetles released at 60 m (Figure 2f). The proportion of latecomers tended to be higher for beetles released at the longest distances. The number of beetles caught from 60 m was low at day 12, negligible at 24 h, and zero at 1 h. The recapture peak for all species was recorded after day 1 for the 2 m release distance. Recapture significantly decreased three to five days after release for all distances up to 20 m (Figure 2). 

Within 24 h from release, most of the recaptured males of all four species had come from within 10 m, although the recapture rate decreased as the release distance increased. The recapture rate decreased at the other distances (Figure 3). The cumulative recapture rate decreased with increasing release distance, following a similar pattern for all four species.

All of the fixed effects included in the ANOVA model (distance, species, and wind direction) were significant (*p* < 0.001), whereas only temperature was significant between covariates. No significant effect was detected for interaction between species and distance, or between species and wind direction. Distance proved to be a major factor in the recapture rate, with a significant reduction trend at the longer distances. From 15 m upwards, the cumulative recapture rate was significantly lower when compared with 10 m or below (Figure 4). 

When distances from 2 m to 10 m were compared, the reduction trend in recapture was the following: 44% for 2 m; 39% for 5 m; 28% for 10 m (Figure 4). After 10 m, no significant difference was detected between the other distances. A higher recapture rate was recorded for *A. litigiosus* and *A. sordidus* than for the other two species. The significant wind direction effect was due to the lower recapture rate for downwind releases (i.e., beetles that had to reach the trap by following the prevailing wind). A one-degree increase in mean temperature caused recapture rate to decrease by 3% (the estimated regression coefficient was −2.61 ± 0.46).

Table 2 reports the results of calculations for maximum sampling range and effective sampling area (r_s_ and α). The best fit linear model for the majority of species was relation r-log Pr (R^2^ > 0.85) for data at 24 h and at day 12 (R^2^ > 0.65). The estimated attraction range and sampling areas of pheromone traps were low (from 53 to 86 m and from 509 to 2602 m^2^, respectively) for all the considered *Agriotes* species, increasing from *A. brevis* (the lowest) to *A. litigiosus* (the highest).

## 4. Discussion

Several attempts have been made to estimate the distance from which male click beetles are attracted to, and caught in, a pheromone trap. Sufyan et al. [28] used the same methods (including the statistical model) described in this paper with two different species (*A. obscurus* and *A. lineatus*), which are major pests, particularly in Central and Northern Europe; Hicks and Blackshaw [27] used a very similar method and statistical model with *A. sputator*, in addition to *A. obscurus* and *A. lineatus*. Our study supplies information about four more species (*A. brevis*, *A. sordidus*, *A. litigiosus,* and *A. ustulatus*), thus contributing to a broader understanding of the attraction range of pheromone traps to the most harmful wireworm species in Europe. Two more species remain to be studied from the list of the most harmful *Agriotes* species stated in the introduction. One is *A. proximus*, which is indistinguishable from *A. lineatus*, leading Staudacher et al. [36] to raise questions about their species status, as the authors found that the two beetles are molecularly indistinguishable. The other is *A. rufipalpis*, whose adults are clearly molecularly [36] and morphologically distinguishable from *A. sordidus*, but both are attracted by the same pheromone [32]. We can thus hypothesize that *A. rufipalpis* behaves similarly to *A. sordidus*. Miller et al. [37] also proposed an interesting approach for studying insect reactions to pheromone sources. It is fairly similar to that used in this paper, but has been used mainly for non-Coleoptera pests: *Cydia pomonella* [38], *Drosophila suzukii* [39], *Halyomorpha halys* [40], and *Lymantria dispar* [41].

The overall recapture rate for the four South-Eastern European species (*A. brevis*, *A. litigiosus*, *A. sordidus* and *A. ustulatus*) was much lower than that found by Sufyan et al. [28] and by Hicks and Blackshaw [27]. The explanation might be that *A. obscurus* and *A. lineatus* are more resistant to disturbance (e.g., collection, movement in vials, handling to mark) and/or that marked beetles in Italy underwent more stress due to higher temperatures. With regard to *A. sputator*, this species resembles *A. brevis* in terms of biological cycle [11,19], taxonomic characteristics [33,42] and sex pheromone compounds. Geranyl butanoate is the sex pheromone compound present in both species, with *A. brevis* also having (E,E)-farnesyl butanoate [31]. Hicks and Blackshaw [27] recaptured an extremely low number of *A. sputator* beetles (less than 5% of those released). In the present trials, the recapture rate for *A. brevis* beetles was lower than for the other species, but much higher than in the trial run with *A. sputator* by Hicks and Blackshaw [27]. With regard to the other two species, data from Sufyan et al. [28] generally agree with the values supplied by Hicks and Blackshaw [27], although they do report a much lower value for the *A. lineatus* estimated sampling area (Table 3). 

Local conditions and/or different population characteristics might be the cause, but we should nevertheless consider that the outputs in Hicks and Blackshaw [27] are less reliable than those from Sufyan et al. [28] due to the lower number of beetles released and to the experiment lasting only one year. Although all of the relevant studies were conducted under different conditions, the following seems valid for all species: (i) the attraction range of pheromone traps and the number of beetles recaptured in open fields are low (the majority within 10 m as shown in this study and previously [27,28,43]) and decrease as the release distances increases; this agrees with an experiment conducted in controlled conditions by Blackshaw et al. [44], who observed that sex pheromones had an attraction range of <5 m in still air (*A. obscurus* only), and a few meters more when air flow was added to the pheromone source; (ii) wind direction may influence attraction range [43,45]; our study achieved lower recapture rates for beetles released downwind (i.e., west of the trap with a prevailing east wind), in contrast to that observed in controlled conditions [44]; (iii) according to previous studies, the majority of beetles are caught from short distances (5–10 m) only within 24 h (Figure 2 and Figure 3); the number of recaptured *A. lineatus* and *A. obscurus* was, however, higher at short release distances [27,28,43] compared with the findings of our study; (iv) it seems that beetles cannot respond to pheromone from 60 meters; captures may therefore depend on movement that can cause the beetles to randomly enter within a range in which they can perceive the pheromone plumes. This assumption is supported by Schallhart et al. [46] on the natural dispersal of *A. obscurus*, with their experiments finding that *A. obscurus* males were able to migrate up to 80 m; (v) maximum sampling range and effective sampling areas varied greatly with species (Table 3), resulting in obvious practical pest management differences. The lowest values were found for *A. brevis*, a mainly crawling species [31]: 53 m and 509 m^2^, respectively (Table 3) about two and four times lower than *A. litigiosus* and *A. obscurus* [28]. It is likely that the greater the propensity for flight, the greater the effective sampling area, since there is a higher probability of a beetle entering the area where pheromone plumes can be perceived.

Many successful cases of mass trapping, including of Coleoptera, have been reported worldwide [47]. One case was in Japan where it was used to control *Melanotus okinawensis* (Elateridae) on sugarcane, with adult densities being reduced by approximately 90% after six years of mass trapping with 10 pheromone traps per hectare [48]. In contrast, a similar study observed no reduction in *Melanotus sakishimensis* abundance [49]. Despite the reports of successful cases, the available data indicate that effective mass trapping of male click beetles with sex pheromone traps would be difficult and sometimes expensive. The first practical attempts using an affordable 40 m trap grid had no effect [50]. In order to be effective, mass trapping should be complete and immediate, as catching males after they have already mated is useless. The results of these and previous trials [27,28] showed that a high male recapture rate takes several days, particularly at >5 m release distances, and that a significant part (about 20%) of the males are not caught [51], even when traps are deployed very densely for mass trapping purposes [52]. This is because a proportion of the males do not react to pheromone plumes, even when they are a short distance from the emitting source [44]. Therefore, the probability that they have already mated before being caught is high, especially for those arriving from longer distances. High and immediate male catches can be achieved only with a dense grid of traps, i.e., 10 meters apart, so that theoretically a male in the field that emerged from the same field, or arrived from outside, will only ever be a maximum of 5 meters from a trap. This means that a huge number of traps needs to be placed per hectare, leading to even higher costs than those reported by Hicks and Blackshaw [27]. Vernon et al. [52] also remark that these costs would currently be unaffordable for an arable crop farmer. Nor can this large-scale trap deployment prevent females entering the treated field from outside, as they are attracted by floral volatiles [53,54,55]. It is also impossible to prevent a portion of the uncaptured males mating with many females, which means that some eggs will always be laid. In conclusion, the practical use of pheromone traps as a mass trapping tool appears to be unfeasible, both technically and financially, making further research on the issue unappealing. The data available might be useful for setting up mating-disruption trials, possibly using low-cost materials embedded with sex pheromone compounds and distributed across large areas. The choice of the most suitable trap may increase mass trapping effectiveness [30]. However, research data on the potential attraction range of sex pheromone traps enable a significant improvement in IPM strategies, as they show how trap use can be optimized for the following purposes: (i) as a monitoring tool (e.g., interpreting capture data, planning widescale monitoring for IPM implementation), and (ii) as attract-and-kill strategies, using entomopathogens as alternatives to chemical pesticides [56]. The information obtained may suggest the most suitable trap grids for attracting and killing most of the beetle populations. This strategy does not theoretically require a 100% catch, or the vast majority of male beetles to be caught in a short space of time, since the killing agent would spread through the population, coming into increasing contact with both male and female adult beetles. Sex pheromones also perform as aggregation pheromones and attract significant numbers of females, as demonstrated for *A. sordidus*, *A. brevis* and *A. ustulatus* [53,54,55,57]. This may further increase entomopathogenic infections in click beetle populations. It therefore appears very promising for future experiments on wireworm population reduction to test the potential of trap grids inoculated with entomopathogen spores based on the maximum sampling range of each species (Table 3), plus sub-multiples or multiples thereof. Successful results would make effective, low-cost wireworm population control tools available, thus providing an alternative to pesticides when IPM procedures have identified a high risk of wireworm damage to arable crops [26].

## Figures and Tables

**Figure 1 insects-12-00733-f001:**
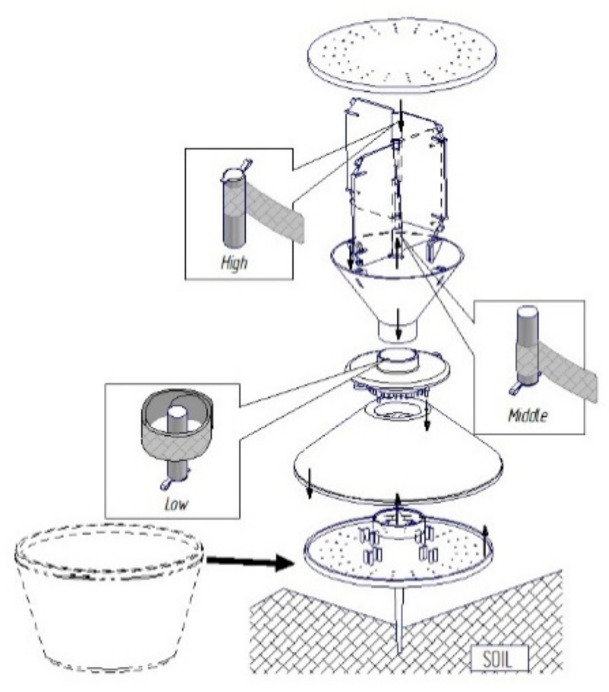
YATLORf trap and the different lure positions.

**Figure 2 insects-12-00733-f002:**
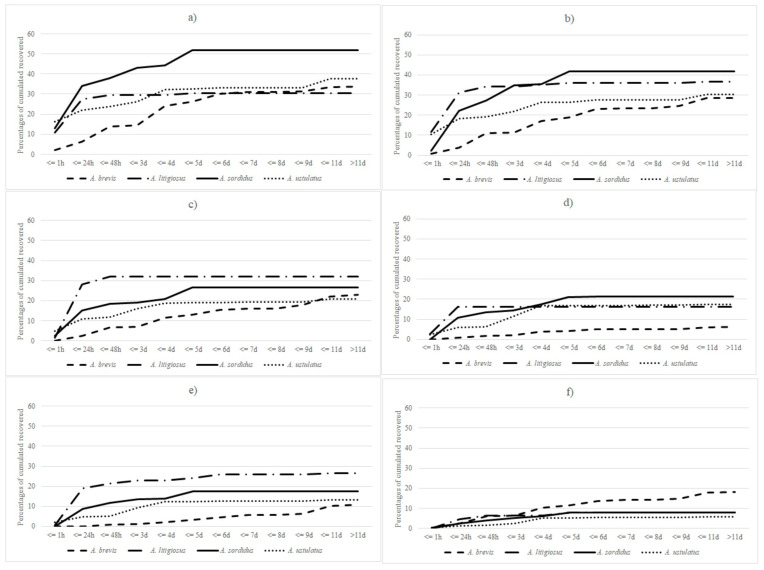
Percentage of cumulative recaptured beetles by time at each release distance: (**a**) 2 m; (**b**) 5 m; (**c**) 10 m; (**d**) 15 m; (**e**) 20 m; (**f**) 60 m.

**Figure 3 insects-12-00733-f003:**
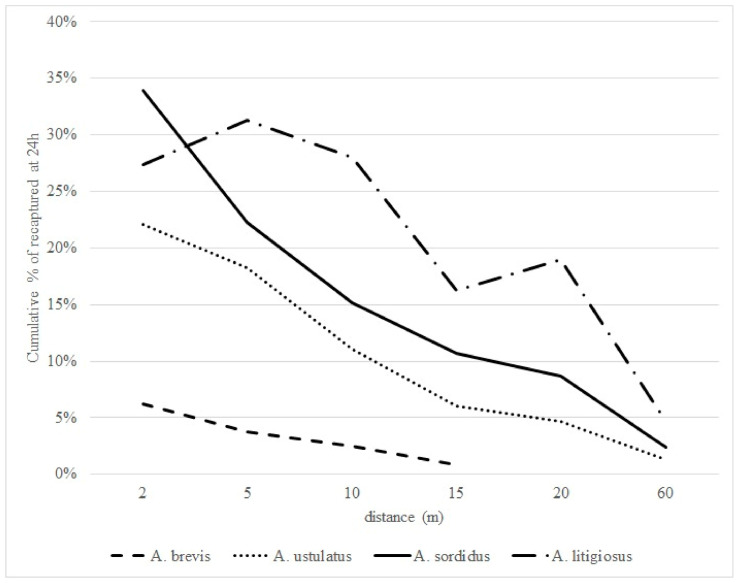
Recapture rate at 24 h at each release distance.

**Figure 4 insects-12-00733-f004:**
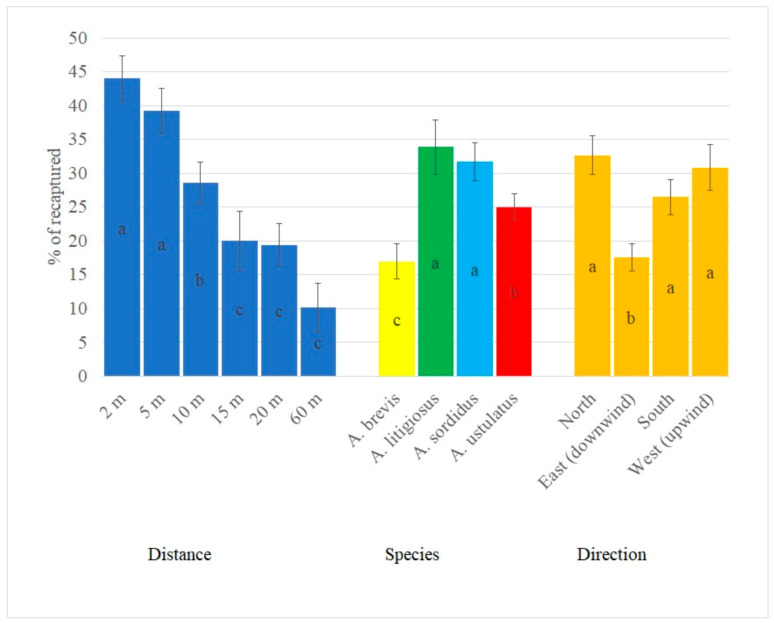
Results of ANOVA: effect of distance from the trap, species, and direction (east is a release position upwind of the trap, with beetles moving downwind towards it; west is a release position downwind of the trap, with beetles moving upwind towards it). Means with the same lowercase letters are not significantly different at *p* < 0.05 (Tukey’s test).

**Table 1 insects-12-00733-t001:** Number of released beetles per year, date of trial, species, wind direction, and distance from trap (released at each point and total). Some environmental (precipitation and mean temperature) and agronomic (crop) details are reported.

Year	Date	Species	Wind Direction	Distance from Trap						Released at Each Point	Total Released	Precipitation (mm)	Temperature °C	Crop
				2 m	5 m	10 m	15 m	20 m	60 m					
2000	23 June	*A. litigiosus*	South			X		X		30	60	0	25	Soybean
	26 June	*A. ustulatus*	East			X			X	100	200	0	20	Bare/sugar beet
	29 June	*A. litigiosus*	South			X		X		30	60	0	21	Soybean
	2 July	*A. litigiosus*	South			X		X		30	60	0	25	Soybean
	7 July	*A. ustulatus*	East			X			X	50	100	0	25	Bare
	12 July	*A. ustulatus*	East			X			X	50	100	0.6	19	Bare
	20 July	*A. ustulatus*	East		X	X		X		10	30	0	21	Bare
2001	1 March	*A. brevis*	South	X	X	X		X	X	50	250	0	3	Bare
	12 March	*A. brevis*	South	X	X	X		X	X	50	250	1.6	10	Bare
	10 April	*A. brevis*	All	X						25	100	0	13	Bare
	14 April	*A. brevis*	All					X		25	100	0	8	Bare
	6 May	*A. sordidus*	North, South	X	X	X		X	X	25	250	2.1	19	Maize
	6 May	*A. sordidus*	East, West	X	X	X				25	150	2.1	19	Maize
	10 May	*A. sordidus*	North, South	X	X	X		X		25	200	0	20	Maize
	12 May	*A. sordidus*	North, South	X	X	X		X		25	200	0	20	Maize
	19 May	*A. sordidus*	North, South		X	X		X		25	150	0	18	Maize
	19 May	*A. sordidus*	East, West		X	X				25	100	0	18	Maize
	6 June	*A. litigiosus*	East, West	X	X	X		X	X	20	200	2.3	16	Maize
	19 June	*A. ustulatus*	North, South	X	X	X		X	X	15	150	0.2	19	Maize
	19 June	*A. ustulatus*	North, South		X	X		X	X	30	240	0.2	19	Maize
	23 June	*A. litigiosus*	East	X	X	X		X	X	20	100	0	22	Soybean
	23 June	*A. ustulatus*	North, South	X	X	X		X	X	15	150	0	22	Maize
	30 June	*A. ustulatus*	North, South		X	X	X	X	X	15	150	3	25	Maize
	13 July	*A. ustulatus*	East	X	X	X	X	X	X	20	120	0	22	Bare
	15 July	*A. ustulatus*	East, West	X	X	X	X	X	X	20	240	0	25	Soybean
	23 July	*A. ustulatus*	East	X	X	X	X	X		20	100	0	23	Soybean
	24 July	*A. ustulatus*	East	X	X	X	X			15	60	0	24	Bare
	3 August	*A. ustulatus*	East	X	X	X		X		15	60	0	26	Bare
2002	5 April	*A. brevis*	All	X	X	X	X	X	X	25	600	0.1	11	Bare
	11 April	*A. brevis*	All	X	X	X	X	X	X	25	600	11.2	8	Bare
	26 April	*A. brevis*	All	X	X	X	X	X	X	25	600	0	17	Bare
	15 May	*A. sordidus*	North, South	X	X	X	X	X	X	50	600	0	20	Bare/Meadow /WW
	15 May	*A. sordidus*	East	X	X	X	X	X	X	75	450	0	20	Bare/Meadow /WW
	18 June	*A. litigiosus*	East	X	X	X	X	X	X	40	240	0	27	Bare/Maize
	24 June	*A. litigiosus*	East	X	X	X	X	X	X	20	120	2.7	27	Bare
	27 June	*A. litigiosus*	East	X	X	X	X	X	X	20	120	0	23	Bare
	1 July	*A. ustulatus*	North, South	X	X	X	X	X	X	25	300	0	21	Bare
	1 July	*A. litigiosus*	East	X	X	X	X	X	X	20	120	0	21	Bare
	4 July	*A. ustulatus*	All	X	X	X	X	X	X	25	600	0	23	Maize
	5 July	*A. ustulatus*	East, West	X	X	X	X	X	X	25	300	0	22	Bare
	5 July	*A. ustulatus*	East, West	X	X	X	X	X	X	25	300	0	22	Maize
	24 July	*A. ustulatus*	North, South	X	X	X	X	X	X	25	300	2	23	Bare
	24 July	*A. ustulatus*	North, South	X	X	X	X	X	X	25	300	2	23	Soybean
2003	30 April	*A. sordidus*	East	X	X	X	X	X	X	40	240	0	15	Bare/WW
	2 May	*A. sordidus*	East	X	X	X	X	X	X	40	240	0	17	Alfalfa/Maize /Soybean/WW
	6 May	*A. sordidus*	East	X	X	X	X	X	X	40	240	0	20	Alfalfa/Maize /Soybean/WW
	8 May	*A. sordidus*	East	X	X	X	X	X	X	40	240	0	21	Bare/WW
	15 May	*A. sordidus*	East	X	X	X	X	X	X	40	240	0	15	Bare/WW
	8 July	*A. ustulatus*	East	X	X	X		X		60	240	0	24	Alfalfa/Maize /Soybean
	14 July	*A. ustulatus*	East	X	X	X		X		60	240	0	24	Alfalfa/Maize /Soybean

**Table 2 insects-12-00733-t002:** Calculated regressions for the relationship between release distance (r) and the probability of recapture P(r) and estimated sampling ranges (r_s_) and areas (α) for considered click beetles at 24 h and at day 12. Best fit regressions are shaded.

Species	N° of Trials		x	r	log(r)	r	log(r)	r_s_ (Sample Range) m	α (Sample Area) m^2^
			y	P(r)	P(r)	log P(r)	log P(r)		
*A. brevis*	7	at 24 h	equation	y = −0.0065x + 0.0946	y = −0.0892x + 0.1083	y = −0.0731x + 0.9602	y = −0.8719x − 0.889	14	16.57
			R^2^	0.70	0.91	0.94	0.92		
		at D12	equation	y = −0.0053x + 0.2798	y = −0.2015x + 0.374	y = −0.0244x + 0.4942	y = −0.7471x − 0.2207	53	509
			R^2^	0.67	0.86	0.91	0.75		
*A. ustulatus*	9	at 24 h	equation	y = −0.0048x + 0.2192	y = −0.2205x + 0.3471	y = −0.0238x−0.6545	y = −0.8257x − 0.2647	46	332
			R^2^	0.45	0.87	0.86	0.95		
		at D12	equation	y = −0.0052x + 0.2989	y = −0.2182x + 0.4141	y = −0.0137x−0.5332	y = −0.4883x − 0.2977	56	975
			R^2^	0.56	0.88	0.77	0.89		
*A. sordidus*	13	at 24 h	equation	y = −0.00052x + 0.272	y = −0.2374x + 0.4055	y = −0.0196x + 0.5413	y = −0.6762x − 0.2339	52	603
			R^2^	0.54	0.97	0.91	0.92		
		at D12	equation	y = −0.0061x + 0.3832	y = −0.2576x + 0.5206	y = −0.0125x − 0.4137	y = −0.4448x − 0.206	63	1583
			R^2^	0.56	0.85	0.79	0.85		
*A. litigiosus*	21	at 24 h	equation	y = −0.0043x + 0.2908	y = −0.16x + 0.3813	y = −0.0136x − 0.4727	y = −0.4582x − 0.2422	68	1363
			R^2^	0.85	0.74	0.93	0.65		
		at D12	equation	y = −0.0039x + 0.3374	y = −0.1353x + 0.407	y = −0.0099x − 0.4329	y = −0.3185x − 0.2816	86	2602
			R^2^	0.59	0.43	0.69	0.44		

**Table 3 insects-12-00733-t003:** Overview of the available estimated sampling ranges and effective sampling areas for pheromone traps used to capture the main click beetle species in Europe and North America.

Number of Days	Sampling Range r_s_ (m)					Effective Sampling Area α (m^2^)				
	1	12	15	30	45	1	12	15	30	45
Species										
*A. lineatus*		55 *	43 **	80 ** 72 *	82 **		1089 *	2588 **	6908 ** 1735 *	6768 **
*A. obscurus*		72 *	38 **	42 ** 95 *	51 **		1518 *	2580 **	2795 ** 2633 *	3636 **
*A. sputator*			25 **	22 **	22 **			1698 **	1335 **	1335 **
*A. brevis*	14	53				16.57	509			
*A. ustulatus*	46	56				332	975			
*A. sordidus*	52	63				603	1583			
*A. litigiosus*	68	86				1363	2602			

* Hicks and Blackshaw [27]. ** Sufyan et al. [28]. All the other figures are results from the present study.

## Data Availability

The data presented in this study are available on request from the corresponding author.

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
