# Peer review of "Assessment of the Attraction Range of Sex Pheromone Traps to Agriotes (Coleoptera, Elateridae) Male Click Beetles in South-Eastern Europe"

_insects, 2021, doi:10.3390/insects12080733_

Round 1

Reviewer 1 Report

In the present study, “Assessment of the attraction range of sex pheromone traps to Agriotes (Coleoptera, Elateridae) male click beetles in Southeastern Europe”, the authors demonstrate the active space of pheromone lures for attracting click beetles. The authors add to the existing body of knowledge by including data for four additional species of click beetle pests, along with estimations of the area over which a pheromone trap might be active.

The manuscript is well-written with appropriate impact for the readers of Insects, and I recommend it for publication with only minor revisions.

Table 1: In the “Crop” column there sometimes appears “Alfa alfa”. Should this read “Alfalfa”?

Line 135: I suggest replacing “The beetles were always released East of the trap” with “The beetles were either released East of the trap”. This allows for the following statement regarding beetles released west of the trap without introducing confusion. The full sentence will read, “The beetles were either released East of the trap, with them moving downwind towards the pheromone source (prevailing wind from the East), or west of the trap, with them moving upwind towards the pheromone source (wind likely taking the pheromone plumes to the beetles).”

Lines 222-223: “The significant wind direction effect was due to the lower recapture rate for downwind releases (i.e. beetles that had to reach the trap by following the prevailing wind).”
I think it would be more accurate to say “upwind releases” because the beetles are being released upwind of the trap, rather than the trap being placed downwind of the beetles – the trap is, after all, already placed when the beetles are released.
This may require you to revisit figure 4. It is easier to think of “downwind” and “upwind” as locations, rather than directions. Beetles released East of the trap are upwind of the lure source, and beetles released west of the trap are downwind of the lure source. To a degree this could be a style choice, but I had to read these sections multiple times to understand what was going on here. Incorrect use of “downwind” and “upwind” can confuse readers, and I encourage the authors to be precise in their language.

Lines 202, 265, 276, and 322: The use of the word “conspicuous” is odd. This word carries a connotation that there is something out of the ordinary here. Consider using the suggestions below.
              Line 202: could you rephrase to describe where there are significant differences?
              Line 265: omit the word “conspicuously”
              Line: 276: do you mean that recapture rates are “higher” within 10 m?

Line 300: omit the word “genus” in the parentheses.

References: The page numbers in the references are separated by both n-dashes and m-dashes. I believe the preferred style for Insects is to use an m-dash. Please replace the shorter n-dashes with the longer m-dashes.

Reviewer 2 Report

This is an important study looking at recapture rates and attraction range of click beetle YATLORf pheromone-baited traps. Click beetles are important crop pests in Europe and the initial hope was to use YATLORf traps for mass trapping in order to reduce the use of general insecticide. After evaluating the results of study presented here, along with the results of previous studies, it became apparent that the traps cannot be used for mass trapping, but are still very useful detection tools and have potential for attract and kill method of control.

The simple summary and abstract have some minor mistakes:

Line 16: I would change concrete to significant

Line 17: change to “at different distances…”

Lines 20 and 31: should be “from short distances”

Line 23: mating by males sounds strange, what other mating is there?

Lines 24, 38, 330 remove pest-species, it should be “only as monitoring tools”

The Introduction is well written.

The methodology section is also well-written and explains experimental design and analysis very well, however, the methodology used to analyze the data is outdated. In the past 10 years, a significant research has been conducted to better understand pheromone attraction and interpretation of trap catches, the methodology has been developed and described in Miller et al. 2010, Miller et al. 2015, Adamset al. 2017 Kirkpatrick et al. 2018 and. 2019, Miller 2020, Onufrieva et al. 2020. I urge the authors to refer to the newer methods and use the up to date terminology. Results section would need to be updated as well.

Specific comments:

Line 152: marked beetles

Line 157: I am very confused, line 152 says traps were checked after 12 and 30-36 days, but here you say trials were carried out at intervals of 7 days.

Lines 188 – 189: this means, that it takes 12 days to achieve converged catch, therefore the 7 day interval would not be sufficient, unless 157 means something else

The discussion session seems to be a little bit too long, I wonder if it could be shortened.

References are missing numbers.

Round 2

Reviewer 2 Report

Approach to L. dispar analysis is very different from original method (Miller et al. 2015) and does not use any graphs, it is only one equation. Although original Miller’s method will indeed not work, the updated method described in Onufrieva et al. 2020 should. Publishing a follow up paper is a great idea and will provide a significant contributions to the field of chemical ecology as well as general ecology.

Additional edits are needed before this paper can be published:

Line 153: you changed text to say both marked and unmarked beetles were identified and counted, but you need to specify that only the marked ones were used for the analysis. If that's not the case, the analysis is incorrect.

Line 159: give a range of kilometers, e.g. 1 – 5 km apart, kilometers sounds too vague for a scientific paper

Figure 2: "cumulated" should be changed to "cumulative"

Line 238: “saw the recapture rate go down” is incorrect, language needs to be adjusted to something like “A one-degree increase in mean temperature caused recapture rate to decrease by 3%”

Overall, nice paper.
